

# How many particles make up a chaotic many-body quantum system?

**Guy Zisling[1], Lea F. Santos[2] and Yevgeny Bar Lev[1★]**

**1** Department of Physics, Ben-Gurion University of the Negev, Beer-Sheva 84105, Israel
**2** Department of Physics, Yeshiva University, New York, New York 10016, USA

★ ybarlev@bgu.ac.il

## Abstract

We numerically investigate the minimum number of interacting particles, which is required for the onset of strong chaos in quantum systems on a one-dimensional lattice with short-range and long-range interactions. We consider multiple system sizes which are at least three times larger than the number of particles and find that robust signatures of quantum chaos emerge for as few as 4 particles in the case of short-range interactions and as few as 3 particles for long-range interactions, and without any apparent dependence on the size of the system.

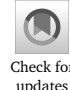
---

## Contents

---

## 1  Introduction

Quantum chaos, especially when caused by particle interactions, has seen a revival in the last decade or so, because it is closely related with topics of high experimental and theoretical in-

terest. It is behind the mechanism of thermalization of isolated many-body quantum systems and the validity of the eigenstate thermalization hypothesis (ETH) [1–3], it explains the heating of driven systems [4,5], it is the main obstacle for many-body localization [6–9], it inhibits long-time simulation of many-body quantum systems [10], it can lead to the fast scrambling of quantum information [11], and it is the regime where the phenomenon of quantum scarring may be observed [12–14].

For systems with a proper semiclassical limit, quantum chaos refers to specific properties found in the quantum domain, when the corresponding classical system is chaotic in the sense of mixing, sensitivity to initial conditions and positive Lyapunov exponents. This correspondence is well established for systems with a few degrees of freedom, such as billiards and kicked rotors, however in the case of systems with many interacting particles, as the ones we are interested in, the correspondence is still lacking due to the challenges involved in their semiclassical analysis [15]. The usual approach is therefore to denote a given system as chaotic if it shows correlated eigenvalues and eigenstates components with similar features to those found in ensembles of full random matrices [16–19].

Most recent studies of quantum chaos in many-body systems are performed for a finite density of particles, but two questions arise: *can quantum chaos occur also at the limit of zero density?* And if so, *how many interacting particles are needed to bring a quantum system to the regime of strong chaos?* These questions are particularly relevant for experiments with cold atoms and ion traps, where the number of particles and also the size of the systems can be controlled. In Ref. [20], by increasing the number of cold atoms step by step, it was experimentally shown that the Fermi sea is formed for as few as four particles. Quantum chaos [18] and thermalization with the appearance of the Fermi-Dirac distribution [21–25] were also obtained with just four interacting particles. More recently, thermalization was studied in systems with 5 particles [26] and quantum chaos was verified again in systems with only 4 particles [27–30], and possibly even with as few as 3 interacting particles [31]. However, it is not entirely clear if other indicators of chaos show similar behaviors, and if the obtained threshold of 4 interacting particles can be changed by the introduction of long-range interactions. These are the questions that we consider in this work.

We focus on spin-1/2 chains with a small number $N$ of excitations and power-law interactions that decay with the distance between the spins. These systems are analogous to systems of hardcore bosons or spinless fermions, such that the number of particles in these cases corresponds to the spin excitations in our models [1]. We find that in systems with short-range couplings, strong chaos emerges already for $N \gtrsim 4$, no matter how large the system size is. While large chains improve the statistics, they do not change our results. We show that long-range interactions can facilitate the transition to chaos and decrease the threshold to only 3 excitations, such that systems with only 3 interacting particles exhibit chaotic properties similar to large interacting systems in the dense limit. This is of particular interest to experiments with ion traps, where the range of interactions can be controlled [32,33], and to studies which explore the generalization of the Lieb-Robinson bound for long-range interacting systems [32–35].

## 2 Model and Chaos Indicators

The spin-1/2 chain that we study is described by the following Hamiltonian

$$\hat{H}_\gamma = \sum_{i=1}^{L-1} \sum_{j=i+1}^{L} \frac{J}{(j-i)^\gamma} \left( \hat{S}_i^x \hat{S}_j^x + \hat{S}_i^y \hat{S}_j^y + \Delta \hat{S}_i^z \hat{S}_j^z \right) + h_1 \hat{S}_1^z + h_{\lfloor L/2 \rfloor} \hat{S}_{\lfloor L/2 \rfloor}^z, \tag{1}$$

---

[1]The analogy is exact only in the limit of nearest-neighbors interactions, as can be seen via the Jordan-Wigner and Holstein-Primakoff transformations

where $\hat{S}_i^{x,y,z}$ are the spin-1/2 operators at lattice site $i$, $L$ is the size of the chain, $J$ is the coupling strength, which we set to be equal to 1, $\Delta$ is the anisotropy parameter, $\gamma$ determines the range of the interactions, whose strengths decay as a power-law with the distance between the spins; and $h_1$ $[h_{\lfloor L/2 \rfloor}]$ is the amplitude of an impurity (defect) placed on site 1 [site $\lfloor L/2 \rfloor$]. The system has open boundary conditions and conserves the total magnetization in the $z$-direction.

We analyze the onset of chaos in sectors with low total $z$-magnetization, where most spins, except a few, point down. Each such sector is characterized by the number of up-spins (excitations). In equivalent models of spinless fermions, this number corresponds to the number of particles. We denote the number of excitations by $N$ and the Hilbert space dimension of the corresponding sector by $\mathcal{D} = \binom{L}{N}$.

## 2.1 Integrable and chaotic points

In the limit of $\gamma \to \infty$ and for $h_1 = h_{\lfloor L/2 \rfloor} = 0$, Eq. (1) describes the XXZ model with nearest-neighbor couplings, which is an integrable model. By adding a small impurity at the border of the chain, we can break the reflection symmetry (parity), but the model remains integrable. Throughout this work we fix $h_1 = 0.11$ and denote this integrable point by $\hat{H}_\infty^0$, where the superscript indicates that $h_{\lfloor L/2 \rfloor} = 0$. To avoid degeneracies, we stay away from the isotropic point and fix $\Delta = 0.55$.

We explore two ways to break the integrability of the XXZ model: by adding an impurity in the middle of the chain, $h_{\lfloor L/2 \rfloor} \neq 0$, and by adding long-range interactions. The fact that the addition of a defect takes the system to the chaotic regime was demonstrated in Refs. [36–39]. Here, without the loss of generality, we choose $h_{\lfloor L/2 \rfloor} = 0.7$ and denote the single-impurity model with nearest-neighbor interactions by $\hat{H}_\infty^{\mathrm{imp}}$. In the absence of the middle-site impurity, integrability is broken by adding interactions between further neighbors, which we do by decreasing the value of $\gamma$. For $h_{\lfloor L/2 \rfloor} = 0$, the system approaches the chaotic domain for $\gamma \lessapprox 5$, but then gets closer to yet another integrable point for $\gamma < 1$. We therefore focus on the interval $1 \leq \gamma \leq 5$.

## 2.2 Indicators of chaos

We employ two indicators of quantum chaos that do not require the unfolding of the spectrum. To detect short-range correlations between the eigenvalues, we use the so-called r-metric, which was introduced in Refs. [40–42],

$$r_\alpha = \min\left( \frac{s_\alpha}{s_{\alpha-1}}, \frac{s_{\alpha-1}}{s_\alpha} \right), \qquad (2)$$

where $s_\alpha = E_{\alpha+1} - E_\alpha$ is the spacing between neighboring eigenvalues of the Hamiltonian. Averaging over all the eigenvalues, $\langle r \rangle \approx 0.39$ for the Poissonian distribution of the spacings, that is often found in integrable models. For chaotic models with real and symmetric matrices, $\langle r \rangle \approx 0.536$. While in our calculations of $\langle r \rangle$, we consider the whole spectrum, it is worth emphasizing that in realistic systems, as the ones we study, chaos develops away from the edges of the spectrum.

Since the eigenstate thermalization hypothesis (ETH) holds due to quantum chaos, we can use the indicators of ETH to detect the transition to chaos. The expectation value of an observable $\hat{O}$ evolves according to

$$O(t) = \langle \Psi | \hat{O}(t) | \Psi \rangle = \sum_\alpha |C_\alpha|^2 O_{\alpha\alpha} + \sum_{\alpha \neq \beta} C_\alpha^* C_\beta e^{-i(E_\beta - E_\alpha)t} O_{\alpha\beta}, \qquad (3)$$

where $C_\alpha = \langle \alpha | \Psi \rangle$ is the overlap between the eigenstate $|\alpha\rangle$ and the initial state $|\Psi\rangle$ of the system and $O_{\alpha\beta} = \langle \alpha | \hat{O} | \beta \rangle$. For sufficiently local observables, ETH builds on two assumptions: that the infinite-time average of $O(t)$, which corresponds to the first term in Eq. (3), coincides with the value of the operator at thermal equilibrium, and that the fluctuations around this value, which are given by the second term in Eq. (3), decrease with system size and cancel out on average. In this work, we focus on the second term, in particular, we investigate the distributions of the off-diagonal elements of the operator, $O_{\alpha\beta}$.

The distribution of the off-diagonal matrix elements, $O_{\alpha\beta}$, in chaotic (thermalizing) systems is Gaussian [43–48], while integrable models have a clear non-Gaussian distribution [44, 47, 49, 50]. The observable that we consider is the magnetization on the impurity-site, $O_{\alpha\beta} = \left\langle \alpha \left| \hat{S}^z_{\lfloor L/2 \rfloor} \right| \beta \right\rangle$, and to assess the chaoticity of the studied systems, we quantify the distance of the distribution of $O_{\alpha\beta}$ from a normal distribution using two measures.

One quantity considered is the kurtosis of the distribution of $O_{\alpha\beta}$,

$$\kappa_{\hat{O}} = \frac{1}{\sigma^4} \left\langle \left( O_{\alpha\beta} - \left\langle O_{\alpha\beta} \right\rangle \right)^4 \right\rangle, \tag{4}$$

where $\langle . \rangle$ indicates the average over all pairs of eigenstates $|\alpha\rangle \neq |\beta\rangle$ and $\sigma$ is the standard deviation of the distribution of $O_{\alpha\beta}$. For Gaussian distributions the kurtosis is $\kappa_{\hat{O}} = 3$. In our plots of the distributions and in our calculations of $\kappa$, we always consider 200 eigenstates with energies closest to the center of the many-body spectrum.

The other metric that we use is

$$\Gamma_{\hat{O}} \left( \omega = E_\beta - E_\alpha \right) = \frac{\overline{\left| O_{\alpha\beta} \right|^2}}{\overline{\left| O_{\alpha\beta} \right|}^2}, \tag{5}$$

which allows to assess the departure from the Gaussianity of the distribution as a function of the energy difference $\omega = E_\beta - E_\alpha$. We extract all the eigenstates that satisfy $(E_\alpha + E_\beta)/2 \in [-0.025\epsilon, +0.025\epsilon]$, where $\epsilon$ is the many-body bandwidth, $\epsilon \equiv E_{max} - E_{min}$, and group these pairs according to their value of $\omega$ in bins of width 0.05 . The overbar in Eq. (5) indicates averaging over the pairs in a given bin. For a Gaussian distribution the value of $\Gamma_{\hat{O}}$ does not depend on $\omega$ and is equal to $\pi/2$ [51].

## 3 Results

We now have all the tools to investigate how the transition to chaos depends on the number of excitations, $N$, for systems with short and long-range interactions.

### 3.1 Short-Range Interactions

We start our analysis by considering the limit of short-range couplings, $\gamma \to \infty$. In the left panel of Fig. 1, we plot $\langle r \rangle$ as a function of the number of excitations for various systems sizes for $\hat{H}^0_\infty$ (triangles) and $\hat{H}^{imp}_\infty$ (circles). As expected, for $\hat{H}^0_\infty$, $\langle r \rangle$ stays very close to 0.39 indicating integrability, with negligible drifts with the system size. On the other hand, for $\hat{H}^{imp}_\infty$, the metric $\langle r \rangle$ increases gradually from an intermediate value between integrability and chaos, $\langle r \rangle \simeq 0.44$, obtained for two excitations, to the chaotic value of $\langle r \rangle \simeq 0.536$ for four or more excitations. The size of the chain does not affect the results.

Figure 2 shows the distributions of the off-diagonal elements of $\hat{S}^z_{\lfloor L/2 \rfloor}$. For chaotic systems without conserved quantities the variance of the off-diagonal matrix elements scales as

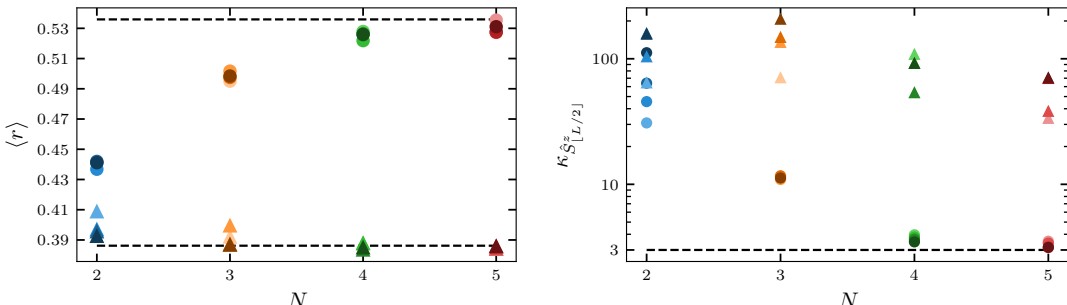

Figure 1: *Left panel*: Quantum chaos indicator $\langle r \rangle$ as a function of the number of excitations $N$ calculated for the integrable model $\hat{H}^0_\infty$ (▲) and the single-impurity model $\hat{H}^{\mathrm{imp}}_\infty$ (●). *Right panel*: same as the left panel, but for the kurtosis $\kappa$ of the distribution of the off-diagonal elements of the middle-site magnetization. Each color represents a number of excitations with the darker shades corresponding to larger system sizes. The system sizes ranges used are: $N = 2, L \in [100, 200]$; $N = 3, L \in [30, 50]$; $N = 4, L \in [22, 28]$ and $N = 5, L \in [16, 21]$.

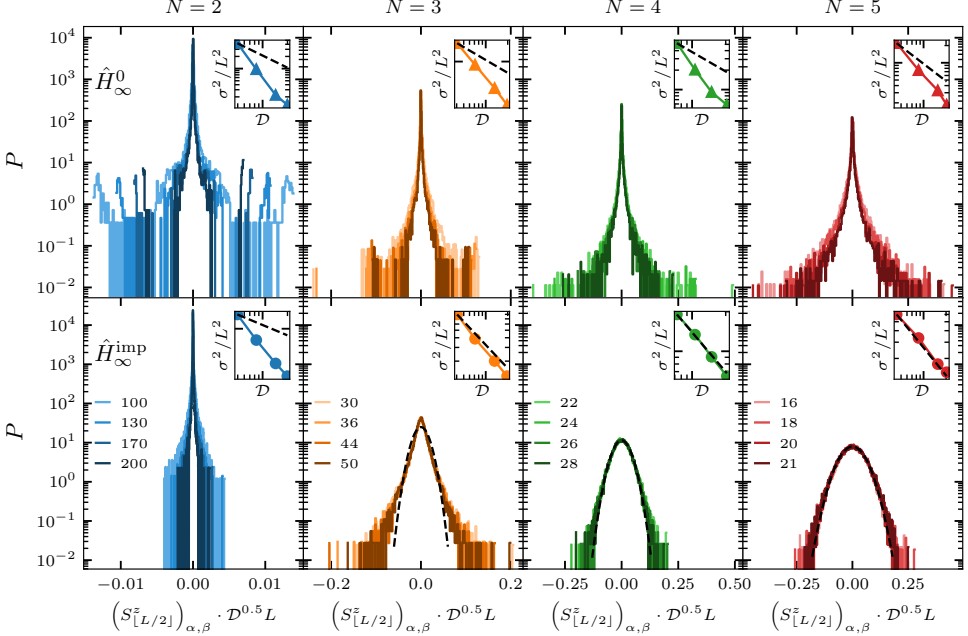

Figure 2: Distribution of the off-diagonal elements of $\hat{S}^z_{\lfloor L/2 \rfloor}$, computed for 200 eigenstates in the middle of the spectrum, for the integrable model $\hat{H}^0_\infty$ (top row) and the single-impurity model $\hat{H}^{\mathrm{imp}}_\infty$ (bottom row) for different number of excitations (different columns). In each panel, larger system sizes are represented by darker colors (see legends). The histograms are scaled by $\mathcal{D}^{0.5}L$. The insets display log-log plots of the scaled variance $\sigma^2/L^2$ against the Hilbert space dimension and the black dashed lines correspond to $\sigma^2/L^2 \propto \mathcal{D}^{-1}$.

$\mathcal{D}^{-1}$, where $\mathcal{D}$ is the Hilbert space dimension. However when conserved quantities are present, the variance decreases slower, as $L^2\mathcal{D}^{-1}$ [2, 50]. Since in our case both the energy and the magnetization are conserved, to plot the distributions of the off-diagonal matrix elements cor-

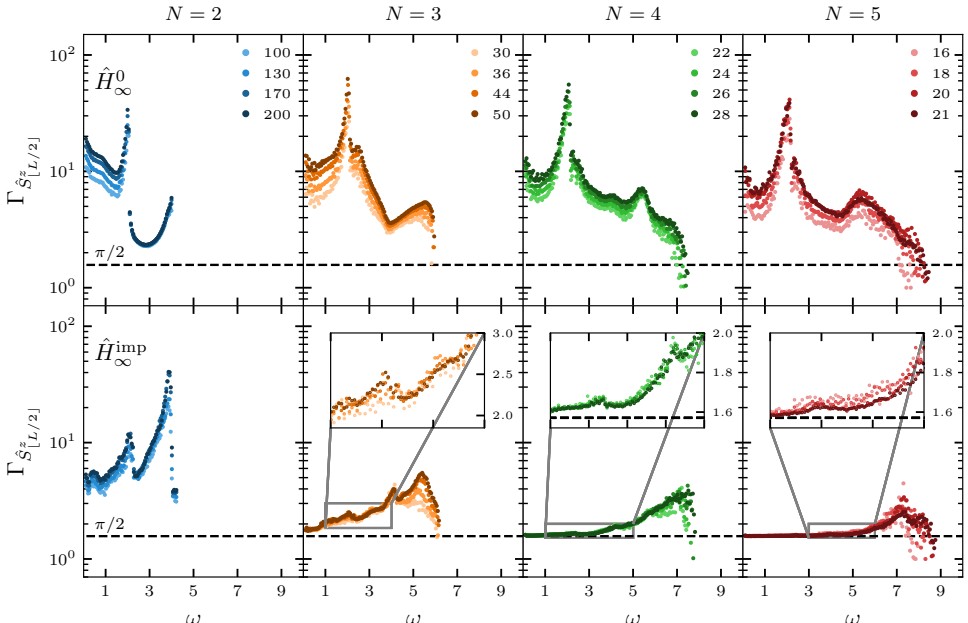

Figure 3: Ratios $\Gamma_{\hat{S}^z_{\lfloor L/2 \rfloor}}(\omega)$ for the integrable model $\hat{H}^0_\infty$ (top row) and the single-impurity model $\hat{H}^{\mathrm{imp}}_\infty$ (bottom row) for different number of excitations (different columns). In each panel, larger system sizes are represented by darker colors (see legends). The insets zoom in into areas of interest and are on a linear scale. See main text for the explanation of how $\Gamma_{\hat{S}^z_{\lfloor L/2 \rfloor}}(\omega)$ was calculated.

responding to different system sizes, such that they will have the same variance, we rescale the values of the off-diagonal matrix elements by the factor $\mathcal{D}^{0.5}L$.

For $\hat{H}^0_\infty$ (top row in Fig. 2), the distributions are visibly non-Gaussian and exhibit a peaked structure for any number of excitations. For $\hat{H}^{\mathrm{imp}}_\infty$ (bottom row in Fig. 2) they are non-Gaussian for $N = 2, 3$, but this changes for $N \geq 4$, which is consistent with our results for $\langle r \rangle$ in the left panel of Fig. 1, where the single-impurity model shows a transition to the regime of strong chaos for 4 or more excitations. For $N \geq 4$ the variance of the off-diagonal matrix elements, as seen in the insets of Fig. 2, decreases as $L^2 \mathcal{D}^{-1}$, as expected for chaotic systems with conserved quantities [2,50]. Notice also that scaled distributions do not show significant dependence on system size, although due to better statistics, the curves become smoother as $L$ increases.

To quantify how close the distributions are to normal distributions, we plot their kurtosis in the right panel of Fig. 1. For $\hat{H}^0_\infty$ (triangles), the kurtosis is much larger than the value which corresponds to a normal distribution, $\kappa = 3$, and it increases with $L$. For $\hat{H}^{\mathrm{imp}}_\infty$ (circles) the kurtosis is close to 3 for $N \geq 4$, converging even closer to 3 as the system size increases.

The behavior of $\langle r \rangle$ in the left panel of Fig. 1 and of the kurtosis in the right panel of Fig. 1 for $\hat{H}^{\mathrm{imp}}_\infty$ shows a very similar trend towards chaos as $N$ increases, namely, as $\langle r \rangle$ approaches its chaotic value 0.536, $\kappa$ approaches 3. We note that these two metrics are very different in nature, since the r-metric has information about the spectrum, while the kurtosis reflects the structure of the eigenstates through the off-diagonal elements of the observable, yet, they provide equivalent information about the onset of quantum chaos.

In Fig. 3, we analyze how close the distributions of the off-diagonal elements of $\hat{S}^z_{\lfloor L/2 \rfloor}$ are to normal distributions, taking into account the energy difference $\omega$. For $\hat{H}^0_\infty$ (top row), the

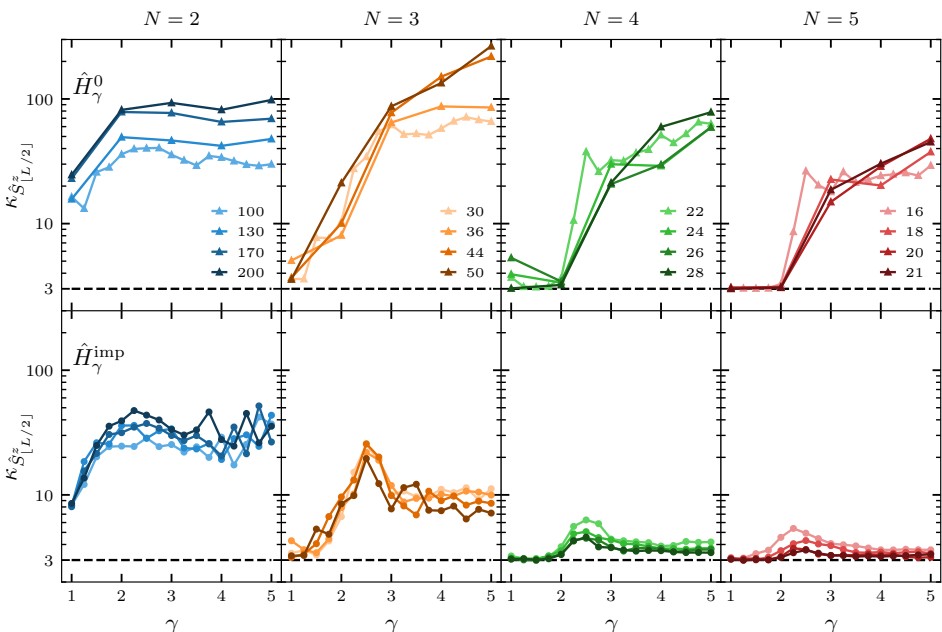

Figure 4: Kurtosis of the distribution of off-diagonal matrix elements computed for 200 eigenstates in the middle of the spectrum for systems with variable ranges of interactions without the middle-site impurity, $\hat{H}_\gamma^0$ (top row), and with a middle-site impurity, $\hat{H}_\gamma^{\mathrm{imp}}$ (bottom row), plotted for different number of excitations (different columns). In each panel, larger system sizes are represented by darker shades (see legends).

value of $\Gamma_{\hat{S}_{\lfloor L/2 \rfloor}^z}$ is larger than the value which corresponds to a normal distribution, $\Gamma_{\hat{S}_{\lfloor L/2 \rfloor}^z} = \pi/2$, for all $\omega$ and it increases as the system size increases. The behavior is very similar for $\hat{H}_\infty^{\mathrm{imp}}$ and $N \leq 3$, although for $N = 3$, as the inset indicates, the deterioration with the system size is less apparent. For $\hat{H}_\infty^{\mathrm{imp}}$ and $N = 4$, we reach a crossing point, where $\Gamma_{\hat{S}_{\lfloor L/2 \rfloor}^z}$ is close to $\pi/2$ for small $\omega$'s and appears to converge to $\pi/2$ with the system size. The improvement with $L$ for $N = 5$ is even more evident and somewhat analogous to the improvement with system size verified in systems with a fixed density [44, 45, 47].

## 3.2 Long-Range Interactions

We now examine how the transition to quantum chaos is affected by the the presence of long-range interactions. Given the similar information obtained with the spectral correlation measure $\langle r \rangle$ and the kurtosis, in Fig. 4 we present only the kurtosis as a function of the coupling range $\gamma$.

For long-range interactions, when $\gamma \sim 1$, the system approaches the chaotic limit, corresponding to $\kappa = 3$, for as few as 3 excitations and this happens for both $\hat{H}_\gamma^0$ (top row) and $\hat{H}_\gamma^{\mathrm{imp}}$ (bottom row). Focusing on the point $\gamma = 1$ there is however no apparent drift towards chaos with the system size.

For $1 \leq \gamma \leq 2$ and $N \geq 4$, it is evident that the system is chaotic and drifts towards chaos as a function of the system size for both $\hat{H}_\gamma^0$ and $\hat{H}_\gamma^{\mathrm{imp}}$. As we leave the region of long-range interactions and $\gamma > 2$, the results for the two Hamiltonians become different, as can

be anticipated, since in this limit the interaction is sufficiently short-ranged that the models become practically indistinguishable from their the local variants (cf. right panel of Fig. 1).

## 4  Discussion

In this work we have numerically studied the quantum chaotic properties of the spectrum and the eigenstates of a prototypical spin-1/2 chain with short and long-range interactions in the limit of a small number of spin excitations. These systems correspond to bosonic or fermionic systems with a small number of particles. Our focus is in the region of the spectrum where quantum chaos is known to develop in systems with many interacting particles, that is, away from the spectrum edges.

We have shown that a large one-dimensional lattice with only four nearest-neighbor-interacting particles or even just three long-range-interacting particles exhibits the same properties of quantum chaos observed in systems with a finite density of interacting particles. Since our results do not appear to depend on the system size, they suggest that the transition to chaos occurs at zero particle density, though further studies are in place, since for four or more excitations, it is challenging to go below a density of 1/6 as the Hilbert space size gets too big for exact diagonalization. Our result is of practical advantage for experiments that have a control over the number of particles and the range of interactions, such as those with ion traps, and which study thermalization and other consequences of many-body quantum chaos. Moreover it offers a simplified scenario for the development of semiclassical analysis of interacting quantum systems.

A natural extension of our work is to search for the differences between interacting chaotic systems at high particle density from those with low and in particular zero density. A specific direction to be considered is the effects of particle statistics, since it has marginal effects for low densities, but not so in the high density limit. Other topics worth investigating include the speed of the evolution, specially, short-time dynamics, where spectral correlations are not yet relevant, and transport behavior. These studies may reveal differences between systems with few and many interacting particles [28], which show similar level statistics and ETH indicators.

## Acknowledgments

This research was supported by a grant from the United States-Israel Binational Foundation (BSF, Grant No. 2019644), Jerusalem, Israel, and the United States National Science Foundation (NSF, Grant No. DMR-1936006), and by the Israel Science Foundation (grants No. 527/19 and 218/19).

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
