# Peer review of "How many particles make up a chaotic many-body quantum system?"

_SciPost Physics, doi:SciPost Phys. 10, 088 (2021)_

## Round 2 · Referee Report · Anonymous (Referee 1) · 2021-2-10

Strengths

  1. Main question and message of the paper are clearly addressed.
  2. Convincing numerical evidence.
  3. Easy to read and follow its ideas.

Weaknesses

None that I could discern.

Report

In their work, the authors investigate the amount of particles (or, equivalently, the amount of
excitations) required to observe the onset of quantum chaos. Indeed, while thermalising and
generic systems have received a renovated interest in recent years due to experimental advances,
most works focus on the Hamiltonian parameters of a given system for which thermalisation and the onset of chaos may be observed and not in the generic qualities of the amount of particles
required. Given that thermalisation in the context of the eigenstate thermalisation hypothesis
(ETH) has become a topic of much interest in recent years, I consider this work to be a rather
timely investigation.

The problem is approached by means of the analysis of statistical indicators of the matrix
elements of a local observable in the eigenbasis of the Hamiltonian and its eigenvalues; namely,
the mean ratio of adjacent energy level spacings and Gaussian statistics of the matrix elements of
the local observable. The latter being a recently found indicator of systems that behave according
to the ETH. In my opinion, the authors' results are well-presented and their numerical analysis
comes across as being convincing.

For the reasons above, I am glad to recommend this work for publication in its current form.
I am certain the work will be appreciated by the community and this journal constitutes a good
venue for it.

The following are only minor comments, which I leave at the discretion of the authors to address if they so desire.

Requested changes

  1. Page 2: It is my impression that reflection symmetry is only present for the XXZ model in the half-filled (zero magnetisation) sub-sector. Parity, however, persists throughout all U(1) sectors and it is indeed broken by the addition of a defect located at the edge. Whether my impression is correct or not, perhaps further clarification regarding these symmetries could be in order.

  2. Page 6: Why is it challenging to go below 1/6 density for four or more particles? Is it due to the size of the Hilbert space or poor statistics?

  • validity: top
  • significance: good
  • originality: ok
  • clarity: top
  • formatting: excellent
  • grammar: perfect

Author:  Guy Zisling  on 2021-03-22  [id 1322]

(in reply to Report 1 on 2021-02-10)

Reply for Report1

We would like to thank the referee for his/her report.

The referee writes:

  1. Page 2: It is my impression that reflection symmetry is only present for the XXZ model in the half-filled (zero magnetisation) sub-sector. Parity, however, persists throughout all U(1) sectors and it is indeed broken by the addition of a defect located at the edge. Whether my impression is correct or not, perhaps further clarification regarding these symmetries could be in order.

Our response:

Parity and reflection symmetry are the same thing. For clarity, we now write:
"we can break the reflection symmetry (parity), but"
Since reflection of the system doesn’t change its magnetization, it exists in all the magnetization sectors. What the referee must actually have meant is the "spin reversal" (the $\pi$-rotation around an axis perpendicular to the z-axis) of all spins. This symmetry indeed exists only at half-filling, but it is also broken by the defect at one edge of the chain. For example, the state 0110 is no longer symmetric to 1001. Since our studies are far from half-filling, we don't need to mention this symmetry.

The referee writes:

  1. Page 6: Why is it challenging to go below 1/6 density for four or more particles? Is it due to the size of the Hilbert space or poor statistics?

Our response:

The Hilbert space size for each sector is given by $\mathcal{D}= L\ choose\ N$, for five excitations and density $\rho\equiv N/L=1/6$, we get $\mathcal{D} = 30\ choose\ 5\simeq 1.4\cdot 10^5$ which is already quite challenging for exact diagonalization algorithms. We have added a short explanation in the paper.

---

## Round 2 · Referee Report · Anonymous (Referee 2) · 2021-3-12

Strengths

1-Clarity in the presentation of results
2-Solid numerical evidence
3-Relevant for state-of-the-art experiments

Weaknesses

  • No weaknesses I can criticize

Report

This work is a study on the effect of low particle density in the appearance of chaos in one dimensional quantum systems. This problem is of great experimental and theoretical interest for the study of quantum many body dynamics. Over the last years much work has been devoted to quantum chaos and thermalization in many body systems but the effects of low number of particles (or excitations) are not yet deeply explored.

The authors present a systematic and methodical examination of gap ratio of consecutive energy levels and eigenstate matrix element statistics of local observables , which are independent and standard measures of the eigenstate thermalization hypothesis. They achieve to see the onset of quantum chaos and identify a minimum number of particles for which the system is chaotic under different measures. Effects of long and short range interactions over such minimum number of particles are identified but not explained, I hope this difference will be addressed beyond numerical evidence in future works. Moreover integrable points in parameter space are also explored accounting for a more complete picture of the crossover from integrability to chaos.

Results are presented in consistent and ordered manner and have exceptional quality. The conclusions drawn from the results are consequent with the ultimate goal of the study. I consider this paper meets all conditions for publication in this journal.

I only have a couple of suggestions for clarifying even further the study. Suggestion A I leave at the discretion of the authors to apply whilst suggestion B I strongly encourage to implement before publication.

Requested changes

A -Include plots of gap ratio and $\Gamma(\omega)$ for the long -range Hamiltonians as supplementary material

B -Clarify why is challenging to go below 1/6 density.

  • validity: high
  • significance: high
  • originality: good
  • clarity: high
  • formatting: excellent
  • grammar: perfect

Author:  Guy Zisling  on 2021-03-23  [id 1327]

(in reply to Report 2 on 2021-03-12)

Reply for Report2

We would like to thank the referee for his/her report.

The referee writes:

A -Include plots of gap ratio and $\Gamma(\omega)$ for the long -range Hamiltonians as supplementary material

Our response:

It is included in the reply.

The referee writes:

B -Clarify why is challenging to go below 1/6 density.

Our response:

See our reply for Report1. We have also added a short explanation in the paper.

Attachment:

Long_range_figures.pdf

---

## Round 3 · List of Changes

- added clarification that by reflection symmetry we mean parity.
- added explanation why it is difficult to go below 1/6 density, due to exponential growth of the Hilbert space size.
- added figures of both Gamma(omega) and the gap ratio in the long range model to the reply to Referee 2.

---

## Editorial Decision

published